# Insomnia and risk of all-cause dementia: A systematic review and meta-analysis

**Mingxian Meng[1,2], Xiaoming Shen**  **[1]\*, Yanming Xie[3], Rui Lan[1], Shirui Zhu[1]**

**1** Encephalopathy Hospital, The First Affiliated Hospital of Henan University of Traditional Chinese Medicine, Zhengzhou, Henan Province, China, **2** The First Clinical Medical College, Henan University of Traditional Chinese Medicine, Zhengzhou, Henan Province, China, **3** Institute of Clinical Basic Medicine, China Academy of Chinese Medical Sciences, Beijing, China

\* sxmdoc@hactcm.edu.cn

## Abstract

### Background

The evidence on the relationship between insomnia and risk of dementia, Alzheimer's disease (AD), and Vascular dementia (VD) is not consistent. We conducted this meta-analysis to examine the evidence for the risk of developing dementia, AD, or VD in patients with all subtypes of insomnia.

### Methods

A comprehensive search of PubMed, Embase, and the Cochrane Library was conducted using the following search strings: 'Insomnia OR Sleep initiation and Maintenance disorders OR Early morning awakening' AND 'Dementia OR Alzheimer's Disease OR Vascular Dementia' AND 'Risk'. Data extraction was done independently by two researchers. Pooled odds ratio (OR) accompanied by 95% confidence interval (CI) were calculated using either a random-effects model or a fixed-effects model. Sensitivity analyses were performed to assess the robustness of the findings. The potential for publication bias was evaluated through Egger's test and Begg's test.

### Results

This meta-analysis included 16 studies with a combined sample size of over 9 million individuals. Pooled analyses revealed a significant association between insomnia and dementia risk (OR = 1.36; 95% CI: 1.01-1.84), with increased risks for AD (OR = 1.52; 95% CI: 1.19-1.93) and VD (OR = 2.10; 95% CI = 2.06-2.14). Subgroup analyses showed no evidence of associations between initial insomnia (OR = 1.01; 95% CI = 0.71-1.31), sleep-maintenance insomnia (OR = 0.88; 95% CI = 0.66-1.17), and early morning awakening (OR = 0.94; 95% CI = 0.83-1.07) with dementia risk. Insomnia patients from Europe (OR = 1.24; 95% CI = 1.14-1.35), Asia (OR = 2.19; 95% CI = 2.06-2.32), and the Americas (OR = 1.05; 95% CI = 1.04-1.07) had varying risks of dementia. Subgroups with less than five years of follow-up (OR = 2.16; 95% CI = 1.81-2.60) exhibited higher dementia risks in insomnia patients, while those with more than five years of follow-up (OR = 1.17; 95% CI = 1.03-1.33) showed a lower risk.

**Data availability statement:** All relevant data are within the paper and its Supporting information files.

**Funding:** The research was financially supported by the China University Industry-Academia-Research Innovation Fund--Huatong Guokang Medical Research Project under grant number 2023HT033 award to Shen Xiaoming, 2024 Henan Provincial Science and Technology Project Award Shen Xiaoming, Fund No. 242102311250 and Construction of the Double First-Class Traditional Chinese Medicine Discipline Project in Henan Province--Cultivating the Innovation Ability of Graduate Students award to Meng Mingxian (HSEP-DFCTCM-2023-8-44, HSEP-DFCTCM-2023-8-27).

**Competing interests:** The authors have declared that no competing interests exist.

## Conclusion

Our meta-analysis reveals that insomnia is linked to the risk of dementia, AD, and VD. These findings suggest that insomnia may significantly contribute to the risk of all-cause dementia, highlighting the importance of early intervention and management of insomnia. Despite our efforts to minimize and explore the sources of heterogeneity, it still remained, and therefore our results should be interpreted with caution.

## Introduction

Dementia is a progressive neurodegenerative disease, with Alzheimer's disease (AD) and vascular dementia (VD) being the two most common types [1]. Every 3 seconds, the global population of individuals with dementia increases by one, doubling every 20 years [2]. Projections indicate that by 2050, the worldwide prevalence of dementia will reach 152 million [3]. The burden imposed by dementia on families, societies, and economies is substantial, with global expenditures reaching up to one trillion dollars annually [4]. Currently, many interventions are attempted for treatment, but the effects are still unsatisfactory [5]. Given these challenges, identifying and addressing modifiable risk factors such as insomnia could be crucial in mitigating the growing public health burden of dementia [6]. According to the Lancet Commission's 2020 report, approximately 40% of dementia cases are attributable to 12 modifiable health-related risk factors [4].

Sleep disorder is arguably a potential risk factor for dementia, albeit not encompassed within the 12 modifiable risk factors mentioned above [4]. Nearly half, or 47%, of individuals aged 65 and older are affected by sleep disorders [7]. Insomnia is the most common sleep disorder [8], with one-third of the world's population affected by insomnia [9]. According to the Diagnostic and Statistical Manual of Mental Disorders (DSM-5), insomnia is defined by difficulty falling asleep, difficulty staying asleep, and early morning awakenings with difficulty returning to sleep [10]. Concurrently, insomnia may manifest with daytime symptoms such as fatigue, diminished energy, and difficulties with cognitive functions, which are frequently the most distressing manifestations experienced by individuals with insomnia [11].

Previous meta-analysis [12–15] have shown the relationship between insomnia and the risk of all-cause dementia, encompassing AD and VD. However, conflicting findings have emerged from other studies [16–18], rendering their conclusions inconclusive. Meanwhile, the precise mechanism by which insomnia contributes to dementia remains unclear. Current hypotheses suggest potential associations with neuronal inflammation [19], disrupted amyloid-beta (Aβ) protein metabolism [20], and cerebrospinal fluid Tau protein level [21].

Given the complex bidirectional relationship between insomnia and dementia, as well as a wealth of novel evidence is continually emerging concerning the association linking insomnia to the risk of all-cause dementia [16,22–28]. It is imperative to comprehensively synthesize all available evidence to quantify the association between insomnia and the risk of all-cause dementia. Therefore, we conducted this meta-analysis to explore the relationship between insomnia and the risk of all-cause dementia.

## Methods

The study adhered to the guidelines outlined in the Preferred Reporting Items for Systematic Reviews and Meta-Analyses (PRISMA) statement [29]. The protocol has been duly registered on the International prospective register of systematic reviews PROSPERO, with the registration number CRD42024502980.

## Data sources

The databases have been searched for this study encompassed PubMed, Embase, and the Cochrane Library, with a cutoff date set at January 15, 2024. We utilized a combination of medical subject headings (MeSH) and free-text keywords to ensure comprehensive coverage of relevant studies. The search terms included combinations of ("Insomnia" OR "Sleep initiation and Maintenance disorders" OR "Early morning awakening") and ("Dementia" OR "Alzheimer Disease" OR "Vascular Dementia") along with ("Risk"). Additionally, Boolean operators were applied to refine the results, and truncation symbols were used where applicable to capture variant word endings. The search was not limited by language or publication year, allowing us to capture studies from diverse geographical regions and timeframes. Furthermore, to maximize the comprehensiveness of the search, we reviewed reference lists of previous meta-analyses and systematic reviews [12–15], identifying any additional studies that may meet our inclusion criteria. Detailed search strategies, including the exact terms and filters used, are provided in Tables S1–S3 in S1 File.

## Eligibility criteria

All included studies met the following criteria: (1) cohort studies or case-control studies based on cohort trials; (2) the correlation between insomnia and the risk of all-cause dementia, Alzheimer's disease (AD), or Vascular dementia (VD) had to be examined; (3) the exposure variable comprises insomnia and its subtypes, encompassing challenges in sleep onset, sleep maintenance, early awakening, and related factors; (4) the outcome is defined as all-cause dementia, AD or VD; (5) the studies included must provide comprehensive risk estimates, including Hazard Ratios (HR), Relative Risks (RR), or Odds Ratios (OR), accompanied by their respective 95% confidence intervals (CIs). In instances where this data is lacking, direct correspondence will be established with the authors to ensure the acquisition of precise risk estimates.

The exclusion criteria: (1) duplicated publications; (2) letters, conference abstracts, and reviews; (3) the outcome is characterized by cognitive impairment that does not progress to dementia or a decline in cognitive abilities; (4) studies that utilized the same database or investigated different aspects of the same population were considered.

## Study selection

Thorough scrutiny of the literature was systematically undertaken by two independent assessors, MMX and SXM, in strict accordance with predetermined criteria for inclusion and exclusion. Initially, duplicate literature was eliminated through a combination of automated processes and manual review by individuals. Subsequently, all literature unrelated to the topic of research was excluded by carefully reviewing titles and abstracts. Finally, the remaining literature underwent a comprehensive approach, involving the downloading of full texts and meticulous reading, with strict adherence to inclusion and exclusion criteria leading to the exclusion of literature that did not meet the specified standards. Throughout the process of study selection, should discrepancies arise between assessors(MMX and SXM), the resolution is achieved through consultation with a third reviewer, XYM.

## Data extraction

The full text of the included studies underwent a comprehensive review, and pertinent data, such as authors, year of publication, study design, and sample characteristics, were initially extracted. A data extraction form was then developed to outline the specific information

to be collected, ensuring both consistency and completeness. Two independent reviewers performed the data extraction to guarantee the accuracy of the collected information through mutual validation [30]. In instances where clarification or additional data were required, authors of relevant studies were contacted to provide necessary details.

## Risk of bias

To evaluate the literature quality, we utilized the Newcastle-Ottawa Quality Rating Scale (NOS) [31]. Independently, MMX and SXM applied the NOS to assess each included study, emphasizing three critical quality dimensions related to selection bias, Comparability Bias, and Outcome Assessment Bias. Studies scoring 7-9 points on the Newcastle-Ottawa Scale were considered high quality a with low risk of bias, 4-6 points indicated medium quality with a moderate risk of bias, and 0-3 points represented lower quality with a high risk of bias. S4 Table exhibited the specific items of the NOS quality assessment form for non-randomized controlled trials. To ensure consistency and reliability in our evaluation, any disagreements between assessors (MMX and SXM) were resolved through discussion or by seeking input from third-party reviewer (XYM).

## Statistical analysis

Adjusted odds ratios (ORs) accompanied by their corresponding 95% confidence intervals (CIs) will be utilized to assess the relationship between insomnia and the risk of all-cause dementia. Heterogeneity was assessed using the $\chi^2$ test and $I^2$ values [32]. A random-effects model was applied where $I^2 > 50\%$[33], and sources of heterogeneity were explored through subgroup analyses and meta-regression. Sensitivity analyses, crucial for ensuring the robustness of findings, will involve systematic exclusion of individual studies followed by rerunning the analysis to validate the overall effect, thereby ensuring that the correlation results are not unduly influenced by any single study [34]. To evaluate potential publication bias, visual inspection of funnel plots and statistical assessment using Egger's test and Begg's test will be conducted [35,36]. This step aims to identify and address potential publication bias, thereby bolstering confidence in the study outcomes. Given the complex nature of insomnia and all-cause dementia, subgroup analyses based on continent, follow-up duration, gender, number of participants, study type, and insomnia diagnostic criteria and dementia diagnostic criteria will be performed. This approach seeks to offer a nuanced and comprehensive exploration of potential variations in associations across distinct subgroups. Random-effects multivariable meta-regression analyses were performed to investigate potential sources of heterogeneity and to assess the impact of moderators, including Continent, follow-up duration, number of participants, study type, insomnia diagnostic criteria, and dementia diagnostic criteria. We also used R to visualize the scores of each part and total score of the NOS scale of the included studies. All statistical analyses were conducted by Stata statistical software (version 14.0) and R4.2.1.

## Ethics statement

All analyses were based on public database; no ethical approval or patient consent was required.

## Results

### Study selection

A total of 1,984 studies were retrieved from the database and 398 duplicates were removed. A further 1,537 studies were excluded by reading the title and abstract. The full texts of the

remaining 53 articles and the additional 5 articles from previous meta-analyses were downloaded and thoroughly examined. Ultimately, 16 studies [16,18,22–28,37–42] that met the inclusion and exclusion criteria were included in this meta-analysis. Specific details of each study that was read in full text and the reasons for exclusion are provided in S5 Table. The flow chart for literature screening is shown in Fig 1.

## Characteristics of included studies

A total of 16 studies were included in this meta-analysis, 11 of which were cohort studies [16,18,22,23,25–27,39–42] and 5 studies [17,24,28,37,38] were case-control studies; with the publication ranging from 1994 to 2023. A cumulative total of 9,016,761 individuals were enrolled in the studies, with 7 studies [16–18,22,25,28,39] encompassing participants from Europe, 4 from Asia [24,37,38,40], and 5 from the Americas [23,26,27,41,42]. Among the studies incorporated in the analysis, 11 were retrospective [16,18,24,27,28,37–42] in nature, while 5 were prospective [17,22,23,25,26]. The follow-up periods spanned from a minimum of

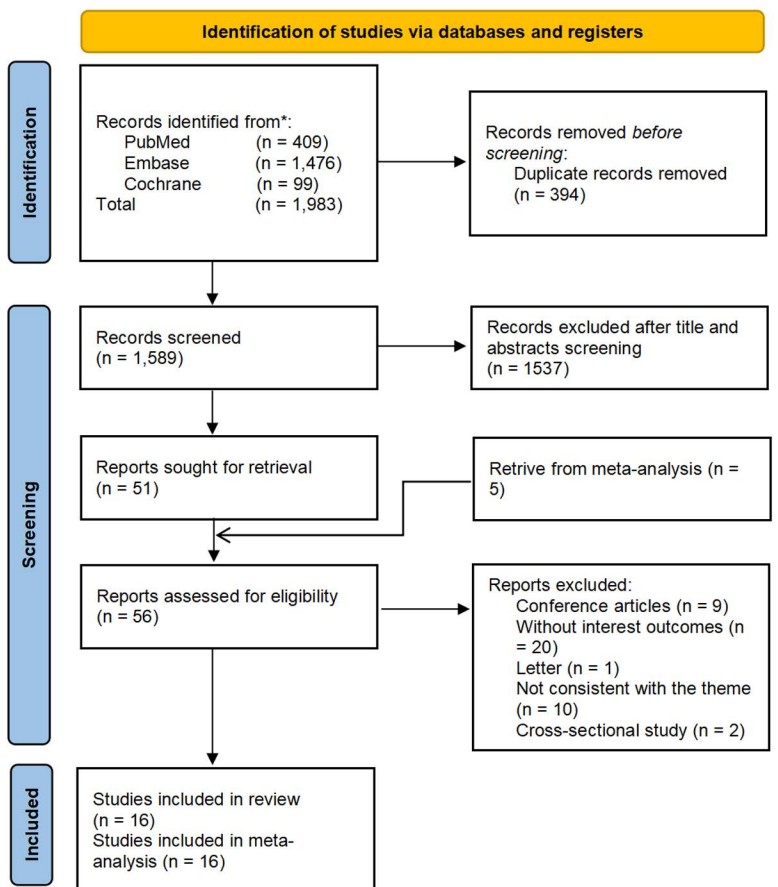

**Fig 1. PRISMA flow diagram illustrating the study selection process for the meta-analysis on the association between insomnia and the risk of All-Cause dementia.** The flow diagram outlines the number of records identified through database searches (PubMed, Embase, Cochrane Library), the number of records screened after removing duplicates, and the number of full-text articles assessed for eligibility. It also shows the number of studies excluded at each stage, with reasons for exclusion provided during full-text assessment. Finally, the diagram highlights the total number of studies included in the qualitative synthesis and those included in the quantitative synthesis (meta-analysis).

3 years to a maximum of 21 years. Self-reported questionnaires emerged as the predominant method for diagnosing insomnia, with a total of 9 studies [17,18,22,23,25–27,39,41] employing this approach. Additionally, 6 studies [24,28,37,38,40,42] utilized the International Classification of Diseases (ICD) diagnostic codes, while 1 study [16] adopted the Diagnostic and Statistical Manual of Mental Disorders (DSM) criteria. The diagnosis of dementia is primarily conducted using ICD coding. All studies, which adjusted for various confounding factors with slight variations across different investigations, presented adjusted risk estimates denoted by HR, OR, or RR, with consistent adjustments for gender and age across all research. The basic characteristics of the included studies are shown in Table 1.

## Quality assessment

The NOS score of all studies included in the Meta-analysis is greater than 7, which indicates the high quality of the included studies. The total score of each study is shown in Table 1, and the detailed scores of each part for selection, comparability, exposure, and outcome are shown in S6 Table. Fig 2 gives a heatmap that illustrates the distribution of bias across the included studies, highlighting the frequency of high, moderate, and low risk of bias in various domains assessed using the Newcastle-Ottawa Scale. Each cell represents the proportion of studies falling into each risk category for a specific domain, with color gradients used to indicate varying levels of bias.

## Insomnia and risk of all-cause dementia

**Risk of all-cause dementia.** A total of 8 studies [16,17,22,27,28,38–40] among 16 studies in this Meta-analysis investigated the relationship between insomnia and risk of all-cause dementia. The pooled analyses with a random effect model showed that insomnia with a high risk of all-cause dementia (OR = 1.36; 95%CI: 1.01-1.84; $I^2$ = 98.7%; $P$ = 0.001). and the sensitive analysis illustrates a robust result present in S1 Fig. Fig 3 presents the forest plot illustrating the association between insomnia and the risk of all-cause dementia, demonstrating a pooled odds ratio of 1.36 (95% CI: 1.01-1.84) with significant heterogeneity ($I^2$ = 98.7%).

**Risk of AD.** Three studies [16,24,37] that met inclusion and exclusion criteria examined the relationship between insomnia and the risk of developing AD. A positive outcome in pool analysis confirms that insomnia was associated with a high risk of developing AD (OR = 1.52; 95%CI: 1.19-1.93; $I^2$ = 65.8%; $P$ = 0.054). As a result of the sensitive analysis displayed in S2 Fig, indicates that the pool analysis of insomnia and risk of AD is robust. Fig 4 demonstrates the forest plot of insomnia and risk of AD, given a pooled odds ratio of 1.52 (95% CI: 1.19-1.93) with significant heterogeneity ($I^2$ = 65.8%).

**Risk of VD.** Two studies [18,37] examined the relationship between insomnia and the risk of VD. Pooled analysis shows a significant association between insomnia and the risk of vascular dementia (OR = 2.10; 95%CI = 2.06-2.14; $I^2$ = 0; $P$ = 0.446). Sensitive analysis reveals robust Meta-analysis results, exhibited in S3 Fig. The forest plot results are depicted in S4 Fig, illustrated the pooled odds ratio of 1.52 (95% CI: 1.19-1.93) with significant heterogeneity ($I^2$ = 65.8%).

**Subtypes of insomnia and risk of all-cause dementia.** Underlying insomnia subtypes, five studies [16,23,25,26,39] examined the effectiveness of the link between initial insomnia and the risk of all-cause dementia. Pooled analysis findings point to no statistically significant association Between Initial Insomnia and risk of all-cause dementia (OR = 1.01; 95%CI = 0.77-1.31; $I^2$ = 83.4%; $P$ = 0.001). The results of the forest plot were shown in S5 Fig. The sensitivity analysis demonstrates the robustness of the meta-analysis results, as presented in S6 Fig.

Apart from this, four studies [16,23,25,26] research on the relationship between sleep-maintenance insomnia and risk of all-cause dementia. After pool analysis, there is no

Table 1. Basic characteristics of the included studies.

| Author | Year | country | Study type | Sample size | Follow-up years | Age (years) | Diagnosis of Insomnia/All-cause dementia | Insomnia type | End-point | Confounders adjusted | NOS scores |
|---|---|---|---|---|---|---|---|---|---|---|---|
| Selbæk-Tungevåg, S et al. | 2023 | Norway | Retrospective cohort study | Total: 7,492 | 11 averages | PID (66.7±6.5) nPID (67.3±6.2) DIS (67.2±6.4) nDIS (67.2±6.2) DMS (67.0±6.3) nDMS (67.3±6.2) EMA (67.7±6.7) nEMA (67.2±6.1) | Insomnia DSM-5 Dementia DSM-5\ MoCA\WLMT | PID, probable insomnia disorder DIS, difficulties initiating sleep DMS, difficulties maintaining sleep EMA, early morning awakenings | All-cause dementia AD | Age, sex, marital status, education, BMI, hypertension, stroke, diabetes, myocardial infarction, heart failure, COPD, sleep apnea, physical activity, smoking, alcohol consumption, sub-scores for depression and anxiety on the HADS, and apolipoprotein E genotype. | 8 |
| Tan, X et al. | 2023 | Swedish | Prospective cohort study | Total: 22,078 | 19 averages | Insomnia (60.6±8.7) nInsomnia (59.8±8.8) | Insomnia (Karolinska Sleep Questionnaire) dementia (ICD-9/ ICD-10) | Insomnia | All-cause dementia | Age, sex, level, depression, social isolation, body mass index, level of physical activity, smoking status, alcohol consumption, hypertension and diabetes. | 8 |
| Lin, W et al. | 2023 | TaiWan-,China | Retrospective case control study | Total: 8,052 | 15 averages | Insomnia (69.01±16.88) nInsomnia (68.95±16.59) | Insomnia (ICD-9-CM) dementia (ICD-9-CM) | Insomnia | AD | sleep disorders, gender, age groups, insured premium, catastrophic illness, diabetic mellitu, hypertension, depression, stroke, dementia, chronic kidney disease, season, location, urbanization level, level of care. | 8 |
| Wong,R et al. | 2023 | United States | prospective cohort study | Total: 6,284 | 10 averages | Insomnia (78.28±8.0) nInsomnia (73.65±6.3) | Insomnia (questionnaire) Dementia (National Health and Aging Trends Study, NHATS algorithm) | Sleep-initiation Insomnia Sleep-maintenance insomnia | All-cause dementia | Sociodemographics and health | 8 |
| Cavaillès, C et al. | 2022 | French | Prospective cohort study | Total: 6,851 | 12 averages | ≥65 | Insomnia (questionnaire) Dementia (DSM-IV) | Difficulty with initiating sleep Difficulty in maintaining sleep Early morning awakening | All-cause dementia | adjusted for study center, sex, mobility, and presence of the APOE-ε4 allele, stratified for the level of education, diabetes mellitus, body-mass index, and cardiovascular disease, depressive status. | 8 |

(Continued)

**Table 1.** (Continued)

| Author | Year | country | Study type | Sample size | Follow-up years | Age (years) | Diagnosis of Insomnia/All-cause dementia | Insomnia type | End-point | Confounders adjusted | NOS scores |
|---|---|---|---|---|---|---|---|---|---|---|---|
| Baek, M.S et al. | 2021 | Korea | Retrospective case control study | Total: 8,390,613 | 7 averages | Insomnia (59.48±11.82) nInsomnia (59.48±11.82) | Insomnia (ICD-10) Alzheimer's disease (ICD-10) vascular dementia (ICD-10) | Insomnia | AD VD | Age and sex | 7 |
| Robbins, R et al. | 2021 | United States | Prospective cohort study | Total: 6,736 | 8 averages | ≥65 | Insomnia (questionnaire) Dementia (Test, Questionnaire) | Difficulty initiating sleep Difficulty falling back asleep Concurrent sleep difficulties (difficulty falling asleep and difficulty falling back asleep) | All-cause dementia | Age, sex, marital status, education and chronic conditions. | 7 |
| Resciniti, N.V et al. | 2021 | United States | Retrospective cohort study | Total: 13,833 | 14 averages | 66.41±9.52 | Insomnia (Questionnaire) Dementia (HRS cognitive score) | Insomnia | All-cause dementia | Age, gender, race, education, body mass index, drinking, smoking status, and chronic disease index | 7 |
| Hoile, R et al. | 2019 | British | Retrospective case-control study | Total: 25,758 | 5 averages | ≥65 | Insomnia (ICD code) Dementia (ICD code) | Insomnia | All-cause dementia | Age, gender, stroke, heart failure, mental illness, sleep apneoa, chronic pulmonry disease, hypnotics sleep apnoea | 8 |
| Hung, C.M et al. | 2018 | Taiwan, China | Retrospective case-control study | Total: 310,458 | 3 averages | Insomnia (47.39±15.69) nInsomnia (47.39±15.69) | Insomnia ICD-9-CM Dementia ICD-9-CM | Primary insomnia | All-cause dementia | Sex, age group, and year of index healthcare use, diabetes, hyperlipidemia, hypertension, coronary heart disease, chronic liver disease, and chronic kidney disease | 8 |
| Sindi, S et al. | 2018 | Sweden (H70 study) | Retrospective cohort study | Total: 437 | 5–9 averages | 70 | Insomnia (Questionnaire) Dementia (DSM criteria) | Initial insomnia in late life Short sleep duration (2-6 hours/night) Long sleep duration (9-11 hours/night) | All-cause dementia | age, sex, education, follow-up time, and study, alcohol consumption, smoking, physical activity, and cohabitant status, cardio/cerebrovascular conditions and hypnotics, hopelessness, APOEε4. | 7 |
| Sindi, S et al. | 2018 | Finland (CAIDE study) | Retrospective cohort study | Total: 703 | 21 averages | 70.2±3.5 | Insomnia (Questionnaire) Dementia (DSM criteria) | Midlife insomnia | All-cause dementia | age, sex, education, follow-up time, and study, alcohol consumption, smoking, physical activity, and cohabitant status, cardio/cerebrovascular conditions and hypnotics, hopelessness, APOEε4. | 7 |

*(Continued)*

Table 1. (Continued)

| Author | Year | country | Study type | Sample size | Follow-up years | Age (years) | Diagnosis of Insomnia/All-cause dementia | Insomnia type | End-point | Confounders adjusted | NOS scores |
|---|---|---|---|---|---|---|---|---|---|---|---|
| Yaffe, K et al. | 2015 | United States | Retrospective cohort study | Total: 179,738 | 8 averages | 66.9 | Insomnia (ICD-9) Dementia (ICD-9) | Insomnia | AD VD Lewy body dementia Other dementia | Age, diabetes, hypertension, myocardial infarction, cerebro-vascular disease, obesity, depression, income tertile, and education. | 9 |
| Chen, P.L et al. | 2012 | Taiwan, China | Retrospective Cohort Study | Total: 33,487 | 3 averages | 65 | Insomnia (ICD-9-CM) Dementia (ICD-9-CM) | Insomnia | All-cause dementia | Age, sex, hypertension, diabetes, hyperlipidemia, and stroke. | 7 |
| Elwood, PC et al. | 2010 | British | Retrospective Cohort Study | Total: 1,985 | 10 averages | Insomnia (61.3) Non-Insomnia (61.9) | Insomnia (Questionnaire) Dementia (AH4, NART, CRT, CAMCOG) | Insomnia | VD | Age, social class, smoking, alcohol intake, BMI, angina, ECG ischaemia and chest pain and National Adult Reading Test | 7 |
| Foley, D et al. | 2001 | United States | Retrospective Cohort Study | Total: 2,166 (male) | 3 averages | Insomnia (76.5±3.7) Non-Insomnia (76.6±3.9) | Insomnia (Questionnaire) Dementia (DSM-III-R) | Insomnia | All-cause dementia | Age, education, apolipoprotein E4 status, Cognitive Abilities Screening Instrument (CASI) score, depressive symptoms, hours of sleep, daytime napping, coronary heart disease, and history of stroke from the baseline examination. | 7 |
| Morgan, K et al. | 1994 | British | Prospective case-control study | Total: 84 | 4 averages | ≥65 | Insomnia (Questionnaire) Dementia (DSM-III-R) | Insomnia | All-cause dementia | Age, sex | 8 |

## Bias Domains Heatmap for Included Studies

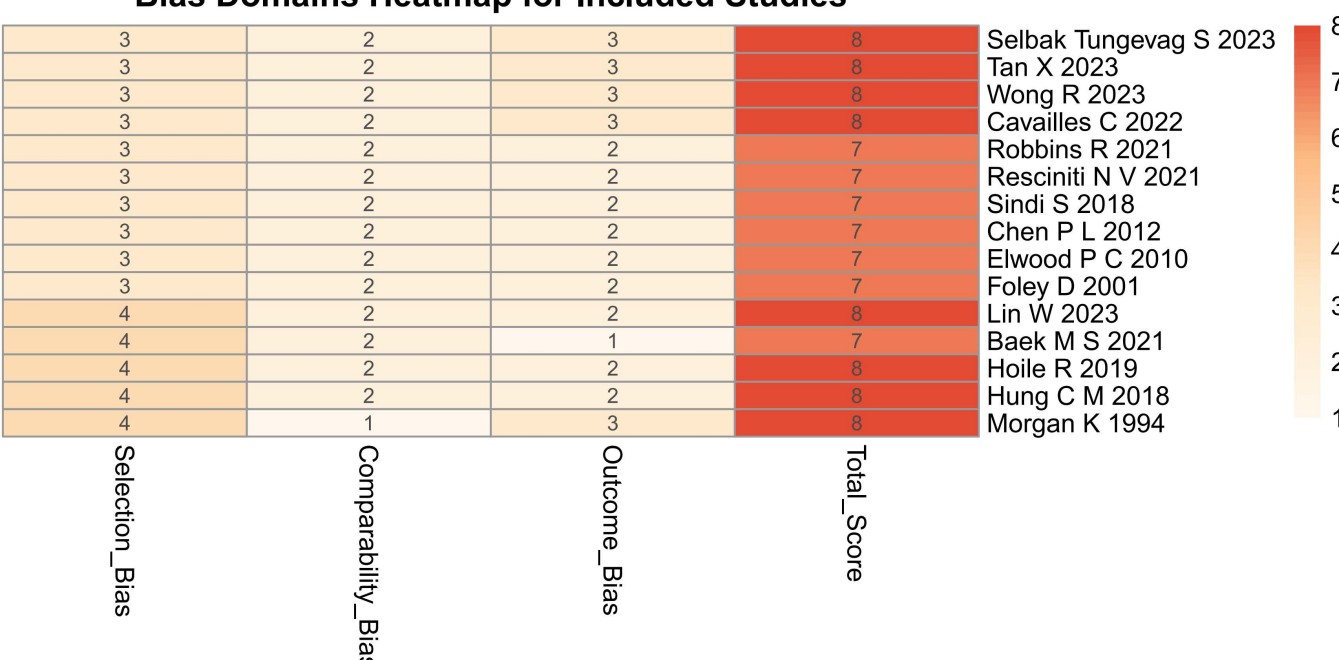

**Fig 2. Bias Domains Heatmap for Included Studies.** The heatmap is color-coded, with darker shades indicating higher risks of bias and lighter shades representing lower risks. The domains evaluated include selection, comparability, and outcome/exposure assessment, with high scores categorized as lower risk.

evidence to suggest that sleep-maintenance insomnia is linked to the risk of developing all-cause dementia (OR = 0.88; 95%CI = 0.66-1.17; $I^2$ = 90.1%; $P$ = 0.001). The forest plot results are displayed in S7 Fig. The robustness of the meta-analysis results is confirmed by the sensitivity analysis, provided in S8 Fig.

Additionally, pool analysis under two studies [16,25] also showed that early morning awakening was not associated with the risk of all-cause dementia (OR = 0.94; 95%CI = 0.83-1.07; $I^2$ = 0; $P$ = 0.939). The forest plot results were depicted in S9 Fig, providing an overview of the individual study estimates and pooled effect size. S10 Fig presents the sensitivity analysis, which verified the stability and reliability of the meta-analysis findings.

### Meta regression

A Multivariate meta-regression analysis was performed to explore the potential sources of heterogeneity in the relationship between insomnia and dementia risk. The model included the following covariates: continent, follow-up duration, number of participants, study type, insomnia diagnosis criteria, and dementia diagnosis criteria. The results showed that no covariates were significantly associated with the effect size, including continent ($p$ = 0.924), follow-up years ($p$ = 0.524), number of participants (p = 0.984), study type ($p$ = 0.527), and insomnia diagnosis criteria($p$ = 0.621), and dementia diagnosis criteria ($p$ = 0.623), did not reach statistical significance. The details of meta-regression results with the risk of All-cause dementia are depicted in Table 2.

### Subgroup analysis

Subgroup analysis was based on the continent of the studies, follow-up years, gender, study type, number of participants, insomnia diagnostic criteria, and dementia diagnostic criteria.

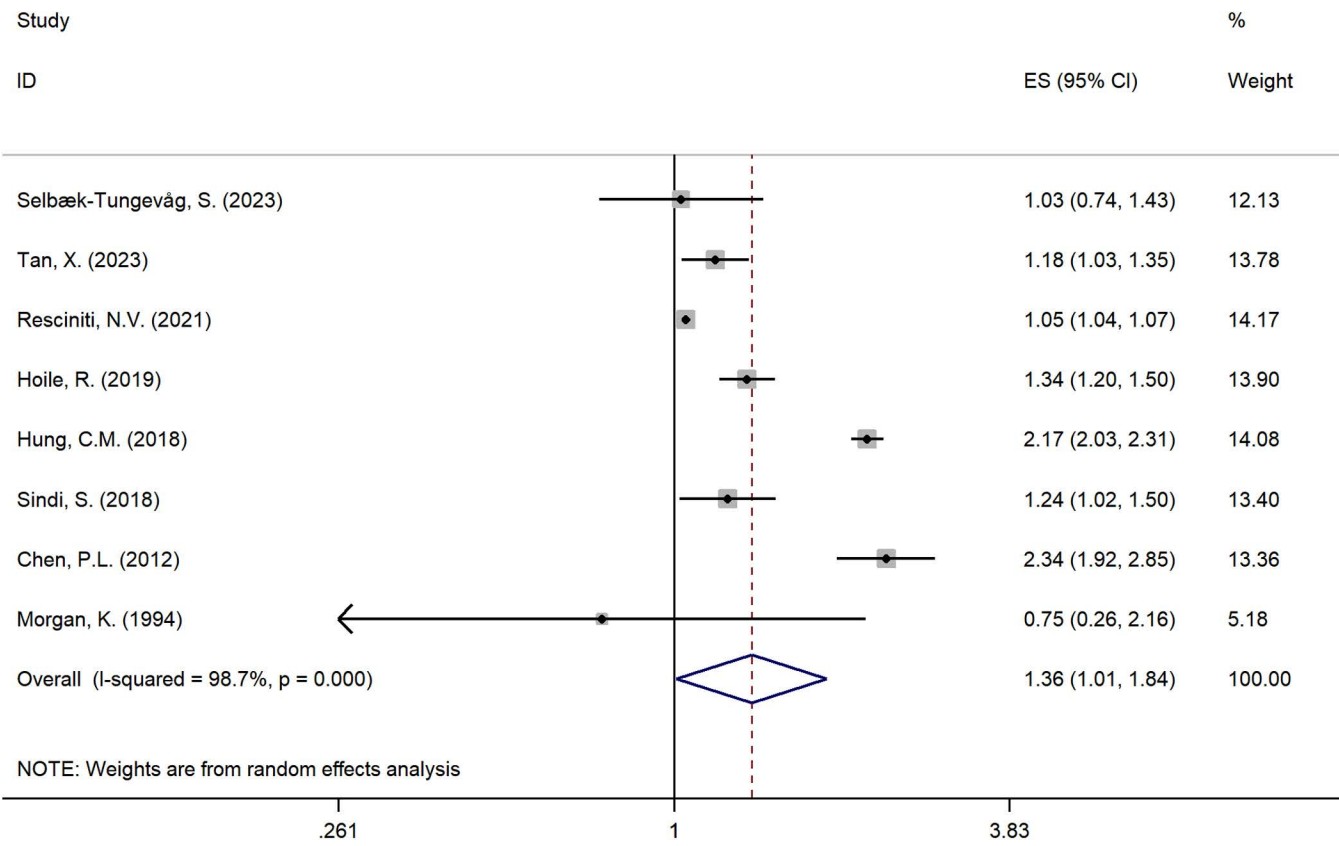

**Fig 3. Sensitive analysis plot of the relationship between Insomnia and risk of all-cause Dementia.** Sensitivity analysis plot showing the impact of each individual study on the overall pooled effect size in the meta-analysis examining the relationship between insomnia and the risk of dementia. Each point represents the recalculated pooled odds ratio (OR) after omitting one study at a time. The horizontal line represents the 95% confidence interval (CI) of the overall pooled OR. The plot demonstrates that the exclusion of any single study does not significantly alter the overall effect size, indicating the robustness of the meta-analysis results.

The pooled results of five studies [16,17,22,28,39] from Europe showed that insomnia increased the risk of all-cause dementia (OR = 1.24), as did the pooled results of two studies [38,40] from Asia (OR = 2.19). In addition, a study [27] from the Americas also confirmed that insomnia is associated with an increased risk of all-cause dementia (OR = 1.05). Under the follow-up time subgroup, the results of the pooled analyses illustrate that follow-up time ≥ 5 years [16,22,27,28] (OR = 1.17), and the follow-up time < 5 years [17,38–40] (OR = 2.16). Subgroups according to gender showed that men with insomnia [40] (OR = 2.39) were more likely to develop all-cause dementia than women with insomnia [40,41] (OR = 1.72). Subgroup analysis based on study type indicated that retrospective studies[16,27,28,38–40] (OR = 1.45) and prospective studies[17,22] (OR = 1.17). According to the number of participants, the subgroup analysis was performed, with the number of participants greater than or equal to 10,000[22,27,28,38,40] (OR = 1.53) and the number of participants less than 10,000[16,17,39] (OR = 1.17). For the subgroup analysis based on insomnia diagnostic criteria, patients were classified into three subgroups: DSM criteria [16], self-report [17,22,27,39], and ICD codes [28,38,40]. The results for the DSM criteria subgroup did not show a significant association with all-cause dementia. ICD codes diagnoses showed a stronger association between insomnia and the risk of All-Cause dementia (OR = 1.89) than self-report (OR = 1.11). Furthermore, subgroup analysis of dementia diagnosis indicated DSM criteria subgroup[16,17,39] had no

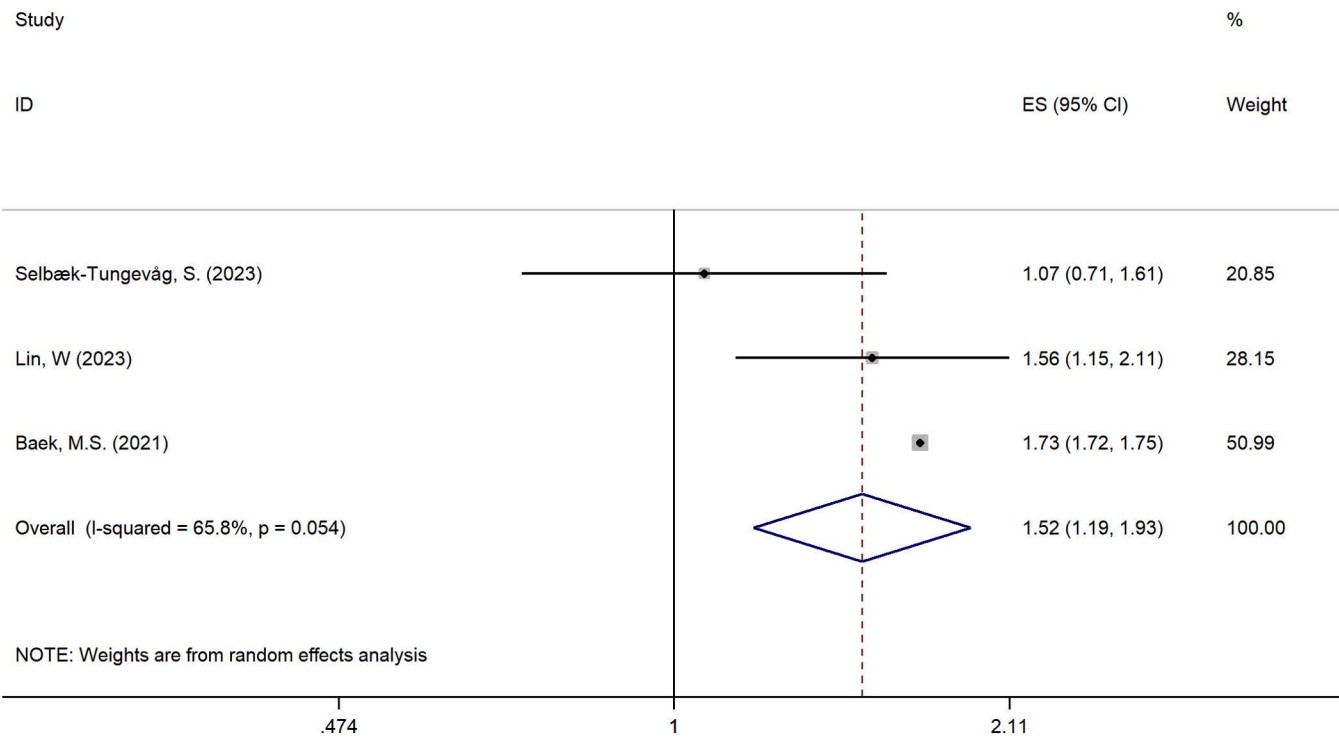

**Fig 4. Forest plot illustrating the association between insomnia and the risk of all-cause dementia.** The plot displays the individual effect sizes (Odds Ratios, ORs) and 95% confidence intervals (CIs) for each study included in the meta-analysis. The solid squares represent the ORs for each study, with the size of the square proportional to the weight of the study in the analysis. The horizontal lines correspond to the 95% CIs. The diamond at the bottom represents the overall pooled OR and its 95% CI, calculated using a random-effects model. An OR greater than 1 indicates an increased risk of dementia associated with insomnia.

**Table 2. Multivariate meta-regression analysis of factors affecting heterogeneity.**

| Variables | Regression Coefficient (95% confidence interval) | Stand Error | $p$ value |
|---|---|---|---|
| Continent | -0.07 (-7.72, 7.58) | 0.60 | 0.924 |
| Follow-up years | 0.56 (-7.12, 8.24) | 0.60 | 0.524 |
| Number of participants | -0.06 (-31.36, 31.24) | 2.46 | 0.984 |
| Study type | -0.50 (-7.46, 6.46) | 0.55 | 0.527 |
| Insomnia diagnostic criteria | -0.38 (-7.43, 6.68) | 0.56 | 0.621 |
| Dementia diagnostic criteria | 0.48 (-8.50, 9.45) | 0.71 | 0.623 |

significant association with all-cause dementia. ICD code diagnoses [22,28,38,40] showed a stronger association between insomnia and the risk of All-Cause dementia (OR = 1.68) than self-report [27] (OR = 1.05). The results of the subgroup analysis for the risk of all-cause dementia in patients with insomnia are summarized in Table 3.

## Publication bias

A visual inspection of the funnel plots revealed no significant publication bias on the studies included in this meta-analysis. Furthermore, validation through Egger's test ($P = 0.204$, $P > 0.05$) and Begg's tests ($P = 0.536$, $P > 0.05$) further confirmed the absence of publication bias. The funnel diagram was presented in Fig 5.

**Table 3. Subgroup analysis for the risk of All-Cause Dementia in patients with Insomnia.**

| Subgroups | Included studies | HR (95%CI) | Heterogeneity | |
|---|---|---|---|---|
| | | | I2(%) | P-value |
| **Continent** | | | | |
| Europe | 5 | 1.24(1.14, 1.35) | 9.5 | 0.352 |
| American | 1 | 1.05(1.04, 1.07) | – | – |
| Asia | 2 | 2.19(2.06, 2.32) | 0 | 0.477 |
| **Follow-up Years** | | | | |
| ≥5 | 5 | 1.17(1.03, 1.33) | 82.9 | 0.001 |
| <5 | 3 | 2.16(1.81, 2.60) | 54.9 | 0.109 |
| **Sex** | | | | |
| Male | 1 | 2.39(1.85, 3.09) | – | – |
| Female | 2 | 1.72(1.02, 1.34) | 0 | 0.941 |
| **Number of participants** | | | | |
| ≥10000 | 5 | 1.53(1.05, 2.23) | 99.3 | <0.001 |
| 10000< | 3 | 1.17(0.99, 1.38) | 0 | 0.448 |
| **Study type** | | | | |
| Retrospective study | 6 | 1.45(1.02, 2.08) | 99.1 | <0.001 |
| Prospective study | 2 | 1.17(1.02, 1.34) | 0 | 0.404 |
| **Insomnia diagnostic criteria** | | | | |
| DSM code | 1 | 1.03(0.74, 1.43) | – | – |
| Self report | 4 | 1.11(1.01, 1.23) | 50.2 | 0.111 |
| ICD code | 3 | 1.89(1.33, 2.68) | 96.5 | <0.001 |
| **Dementia diagnostic criteria** | | | | |
| DSM code | 3 | 1.17(0.99, 1.38) | 0 | 0.448 |
| Self report | 1 | 1.11(1.01, 1.23) | – | – |
| ICD code | 4 | 1.89(1.33, 2.68) | 97.1 | <0.001 |

## Discussion

### Main findings

Our comprehensive analysis revealed that insomnia is significantly associated with an elevated risk of all-cause dementia, as well as AD and VD. Specifically, the risk of dementia increased by 1.36-fold, the risk of AD by 1.52-fold, and the risk of VD by 2.10-fold. However, the subtypes of insomnia, sleep initiation difficulties, sleep maintenance disorders, and early morning awakening were not associated with dementia.

### Comparison with previous studies

The association between insomnia and the risk of all-cause dementia, AD, and VD has yielded conflicting findings in previous research [12,14]. A previous study [12] showed that insomnia increases the risk of all-cause dementia, which is consistent with the results of our meta-analysis. Unfortunately, their study did not analyze the relationship between insomnia and the risk of developing AD and VD. And this meta-analysis included only 5 studies, which may reduce statistical efficacy. In addition, gender confounders were not controlled for. Our study included the most recent and larger cohort studies and controlled the quality of the included studies more strictly by including only cohort studies and case-control studies. At the same time, we tightly controlled for gender factors and did not include studies with only female

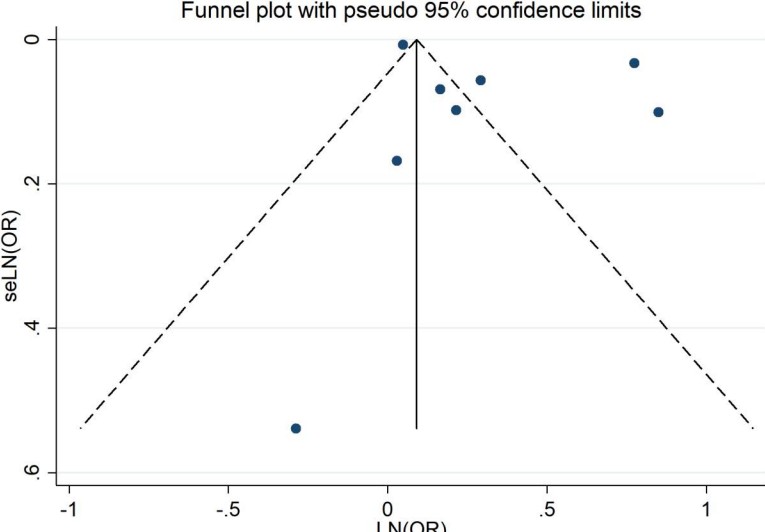

**Fig 5. Funnel plot assessing potential publication bias in studies examining the association between insomnia and the risk of all-cause dementia.** Each point represents an individual study included in the meta-analysis. The x-axis shows the effect size (Odds Ratios, ORs), and the y-axis represents the standard error of the log OR, a measure of study precision. Studies with larger sample sizes and greater precision appear toward the top of the plot, while smaller, less precise studies are near the bottom. Symmetry of the plot indicates the absence of publication bias, whereas asymmetry suggests potential bias or heterogeneity. The vertical dashed line represents the pooled effect size from the meta-analysis.

or male participants in the pooled analysis of insomnia and all-cause dementia risk. Our meta-analysis also found that insomnia was associated with an increased risk of AD and VD, which no meta-analysis had examined before.

While our findings align with some previous studies, the substantial heterogeneity observed underscores the complexity of the relationship between insomnia and dementia. This variability suggests that further research is needed to identify specific population sub-groups or methodological factors that may influence the strength of this association.

### Interpretation of findings

Drawing upon the findings of this meta-analysis and integrating them with existing evidence, we posit that insomnia is linked to a heightened risk of all-cause dementia, AD and VD. However, the precise mechanisms underlying this association remain elusive.

**Amyloid-beta clearance and insomnia.** One of the key mechanisms linking insomnia to an increased risk of Alzheimer's disease (AD) is the impaired clearance of amyloid-beta (Aβ), a hallmark of AD [43]. During sleep, the brain's glymphatic system is more active, facilitating the removal of metabolic waste products like Aβ [20,44,45]. Studies have shown that Aβ clearance is significantly reduced during wakefulness, leading to its accumulation over time [20]. Chronic sleep deprivation, which often accompanies insomnia, may exacerbate this process, resulting in increased Aβ deposition in the brain [46]. This accumulation is believed to accelerate the onset and progression of AD, suggesting that sleep plays a critical role in neuroprotection.

**Tau protein abnormalities and sleep disruption.** In addition to Aβ, abnormalities in Tau protein are another significant factor in AD pathogenesis [47]. Disrupted sleep has been shown to elevate Tau levels in both animal models and human cerebrospinal fluid

(CSF), which may drive neurodegenerative changes. The study by Holth and his colleagues [21] confirmed that the sleep-wake cycle regulates the level of Tau protein in interstitial fluid, which increased by about 90% during wakefulness and by about 100% during sleep deprivation in mice, and by more than 50% during sleep deprivation in human cerebrospinal fluid (CSF), compared with the level of the protein during sleep. Given that Tau protein aggregates are a key feature of AD, chronic sleep disturbances may amplify Tau pathology, accelerating cognitive decline. The study by Rothman [48] further validated this in an Alzheimer's disease mouse model, where sleep restriction (6 hours per day for 6 weeks) exacerbated memory deficits and increased Aβ and Tau levels in the cortex compared to controls. This provides further evidence that sleep deprivation can significantly contribute to both Aβ and Tau accumulation, highlighting the dual impact of sleep on AD pathology.

**Inflammation, vascular dysfunction, and insomnia.** The association between insomnia and vascular dementia (VD) appears to be even stronger than that with AD, suggesting that vascular mechanisms may also be crucial. Chronic sleep loss is known to activate inflammatory pathways, such as nuclear factor kappa-B (NF-κB) and activator of transcription (STAT) family proteins, which can lead to vascular damage and increased risk of cerebrovascular disease [49,50]. Additionally, insomnia has been associated with white matter hyperintensities, which are markers of cerebrovascular dysfunction and are commonly observed in patients with VD [51]. This vascular pathway may explain the particularly strong link between insomnia and VD observed in our analysis, indicating that sleep disturbances could exacerbate both neurodegenerative and vascular pathways contributing to dementia.

**Subgroup analyses findings.** Notably our subgroup analysis based on continent showed that Asian populations had a 2.19-fold increased risk of dementia after insomnia, followed by Europe 1.24-fold and the Americas 1.05-fold. We found that our heterogeneity decreased after subgroup analysis, so we hypothesize that this may be one of the sources of heterogeneity in this meta-analysis. We hypothesize that this phenomenon may be related to genetic predisposition, cultural differences, or variations in healthcare systems. A large case-control study [52] across multiple racial and ethnic groups confirms that the Apolipoprotein E4 (apoE4) genotype is a significant risk factor for late-onset AD, with its risk varying by race, sex, and ancestry. Specifically, the risk associated with apoE4 was higher in East Asians (OR, 4.54) than White (OR, 3.46). These findings emphasize that differently apoE4 status across populations, potentially explaining why different racial and ethnic groups show varying risks of AD. Culture and healthcare systems play a key role in cognitive ability and dementia incidence. East Asian cultures exhibit notable differences in their perceptions of sleep compared to Western cultures. For instance, a study [53] comparing Japanese and European Canadian participants found that the Japanese group perceived a weaker connection between sleep and physical health. Additionally, they reported a significantly shorter ideal sleep duration. These cultural attitudes toward sleep may influence the frequency and severity of cognitive decline associated with insomnia, as reduced sleep duration and a lack of emphasis on its health benefits could exacerbate the long-term impact of sleep disorders. Access to dementia care also differs greatly. In low-to-middle income countries of Asia, dementia is often under-recognized, and healthcare systems are not equipped to address it efficiently [54]. This contrasts with European countries that have more robust healthcare systems, where dementia is recognized earlier, and patients have better access to preventive care for conditions like insomnia [55]. This cultural and healthcare systems context may partly explain regional differences in how insomnia affects the risk of All-Cause dementia. Furthermore, we found that insomnia patients with a follow-up time greater than five years had a lower risk of dementia by subgrouping whether the follow-up time was greater than five years. However, dementia is a long-duration disease especially AD. In six cohorts comprising a total sample of 3,268 individuals, Vermunt

observed that among 70-year-old patients with preclinical Alzheimer's disease (AD), the duration of preclinical AD was 10 years, prodromal AD was 4 years, and dementia was 6 years [56]. so we are more likely to believe that the results of the subgroup analyses with more than five years of follow-up are more realistic. Subsequently, our gender subgroup analysis found that men were at greater risk of developing dementia than women. Although women report a greater incidence of insomnia [57], studies [58] have found that men tend to have more severe outcomes after the onset of insomnia. Moreover, it is reported that the combined incidence of VD and AD is greater in men than in women [59], which may contribute to the higher risk of dementia in men with insomnia. The subgroup analysis based on sample size revealed contrasting results. Studies with a sample size of Greater than or equal to 10,000 indicated a significant association between insomnia and all-cause dementia, whereas studies with fewer than 10,000 participants showed no such association, suggesting potential variability in the observed effects depending on study size. Finally, we did subgroup analyses based on insomnia diagnostic criteria demonstrated that the risk of dementia associated with insomnia was higher in studies using ICD codes for diagnosis compared to those using self-reported diagnoses. This suggests that the method of diagnosis may influence the strength of the observed association between insomnia and risk of dementia. Although most studies diagnose insomnia using questionnaires, such as the National Health and Nutrition Examination Survey (NHANES) in the United States and the UK Biobank, questionnaires may introduce bias due to participant forgetfulness or deliberate concealment. This could potentially weaken the observed association between insomnia and the risk of all-cause dementia.

## Limitations and prospection

To our understanding, this systematic review represents the most extensive and thorough examination of dementia incidence among individuals with insomnia to date. We used a random-effects model to combine all effect sizes. Sensitivity analyses were performed to verify the robustness of the findings. We also used subgroup analyses and meta-regression to explore the sources of heterogeneity. Moreover, the generally high quality of the studies included in our review provides a level of confidence in the results. Nonetheless, our study does have limitations. Clinical heterogeneity and methodological heterogeneity due to differences in demographics and experimental designs are the main sources of statistical heterogeneity in this meta-analysis and cannot be completely avoided. Although we did not identify possible sources of heterogeneity, we speculate that a key limitation is the reliance on self-reported insomnia, which could lead to recall bias or misclassification. Additionally, the considerable heterogeneity observed suggests that other unmeasured factors could influence the outcomes. Therefore, the results of our meta-analysis should be interpreted with caution. Future research should prioritize the use of standardized diagnostic criteria for insomnia and incorporate objective sleep measures to reduce heterogeneity. Long-term follow-up studies, particularly those exceeding five years, are necessary to fully capture the chronic impact of insomnia on dementia risk. Additionally, large-scale, multi-center studies adjusting for key confounders will help identify at-risk populations and clarify the role of insomnia as a modifiable risk factor for dementia.

## Conclusion

This meta-analysis found that insomnia is associated with increased risk of all-cause dementia, as well as AD and VD. Our findings indicate that addressing insomnia through early intervention may be a critical component in reducing the risk of dementia, including AD and VD. This highlights the importance of incorporating sleep assessments into routine clinical practice for at-risk populations.

## Supporting information

**S1 Fig. Sensitive analysis plot of the relationship between Insomnia and risk of all-cause dementia.**
(TIF)

**S2 Fig. Sensitive analysis plot of the relationship between insomnia and risk of AD.**
(TIF)

**S3 Fig. Sensitive analysis plot of the relationship between insomnia and risk of VD.**
(TIF)

**S4 Fig. Forest plot of the relationship between insomnia and risk of VD.**
(TIF)

**S5 Fig. Forest plot of the relationship between initial insomnia and risk of all-cause dementia.**
(TIF)

**S6 Fig. Sensitive analysis plot of the relationship between initial insomnia and risk of all-cause dementia.**
(TIF)

**S7 Fig. Forest plot of the relationship between sleep-maintenance insomnia and risk of all-cause dementia.**
(TIF)

**S8 Fig. Sensitive analysis plot of the relationship between sleep-maintenance insomnia and risk of all-cause dementia.**
(TIF)

**S9 Fig. Forest plot of the relationship between early morning awakening and risk of all-cause dementia.**
(TIF)

**S10 Fig. Sensitive analysis plot of the relationship between early morning awakening and risk of all-cause dementia.**
(TIF)

**S1 File. Supplementary Table S1-S3 Detailed search strategies. Table S1:** PubMed. **Table S2:** Embase. **Table S3:** Cochran Library.
(DOC)

**S4 Table. NOS quality assessment form for non-randomized controlled trials.**
(DOC)

**S5 Table. Reasons for excluded studies.**
(DOC)

**S6 Table. The quality assessment of cohort and case-control studies.**
(DOC)

## Author contributions

**Conceptualization:** Mingxian Meng, Xiaoming Shen.

**Data curation:** Mingxian Meng, Rui Lan, Shirui Zhu.

**Formal analysis:** Mingxian Meng.

**Investigation:** Mingxian Meng.

**Methodology:** Mingxian Meng, Xiaoming Shen.

**Resources:** Rui Lan, Shirui Zhu.

**Software:** Mingxian Meng.

**Supervision:** Yanming Xie.

**Writing – original draft:** Mingxian Meng.

**Writing – review & editing:** Mingxian Meng, Xiaoming Shen.

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
