## [Decision Letter · Decision Letter 0]

8 Sep 2024

PONE-D-24-20907Insomnia and Risk of All-Cause Dementia: A Systematic Review and Meta-AnalysisPLOS ONE

Dear Dr. shen,

Thank you for submitting your manuscript to PLOS ONE. After careful consideration, we feel that it has merit but does not fully meet PLOS ONE’s publication criteria as it currently stands. Therefore, we invite you to submit a revised version of the manuscript that addresses the points raised during the review process.

We look forward to receiving your revised manuscript.

Kind regards,

Rishab Gupta, MD

Academic Editor

PLOS ONE

Journal Requirements:

4. In the online submission form, you indicated that [The datasets used during the study are available from the public databases and corresponding author on reasonable request.]. 

6. As required by our policy on Data Availability, please ensure your manuscript or supplementary information includes the following: 

Additional Editor Comments:

Dear Authors,

I have reviewed your manuscript and also noted the reviewers' comments. I recommend that you make revisions to the manuscript before it can be published.

Reviewers' comments:

Reviewer's Responses to Questions

**Comments to the Author**

1. Is the manuscript technically sound, and do the data support the conclusions?

Reviewer #1: Partly

Reviewer #2: Partly

2. Has the statistical analysis been performed appropriately and rigorously? 

Reviewer #1: I Don't Know

Reviewer #2: Yes

3. Have the authors made all data underlying the findings in their manuscript fully available?

Reviewer #1: Yes

Reviewer #2: Yes

4. Is the manuscript presented in an intelligible fashion and written in standard English?

Reviewer #1: No

Reviewer #2: Yes

5. Review Comments to the Author

Reviewer #1: The article fails on many levels to be considered for publication. For instance, the risk and domains of bias are not elaborately explained. The discussion section also appears to be compromised. The heterogeneity in the means of diagnosis of insomnia is another source of concern; was heterogeneity within limits to allow for a meta-analysis? Were all studies included for systematic review as well as for meta-analysis?

The figures are often given as a passing reference and not adequately explained.

Reviewer #2: Review Comments to the Author:

Manuscript Title: "Insomnia and Risk of All-Cause Dementia: A Systematic Review and Meta-Analysis"

1. Overview: The manuscript titled "Insomnia and Risk of All-Cause Dementia: A Systematic Review and Meta-Analysis" addresses a pertinent issue in the field of sleep disorders and neurodegenerative diseases. The study aims to elucidate the association between insomnia and the risk of developing dementia, including Alzheimer's disease (AD) and Vascular dementia (VD). The manuscript is well-structured, and the research question is clearly articulated. The authors have conducted an extensive and systematic literature search and have utilized appropriate statistical methods to synthesize the findings from multiple studies.

2. Technical Soundness and Data Support: The manuscript is technically sound in many respects, and the data presented do support the authors' conclusions to a certain extent. The meta-analysis incorporates 16 studies, involving over 9 million individuals, which is commendable. The inclusion criteria are well-defined, and the studies included in the analysis are generally of high quality, as indicated by the Newcastle-Ottawa Quality Rating Scale (NOS) scores.

However, there are some concerns that need to be addressed:

Heterogeneity: One of the major concerns is the significant heterogeneity observed in the analysis, particularly in studies with shorter follow-up periods (less than five years). The high I² values in several analyses suggest that there is substantial variability among the studies, which may not be fully accounted for by the random-effects model used. The authors have performed sensitivity analyses to assess the robustness of their findings, but the persistence of heterogeneity indicates that the results should be interpreted with caution. It would be beneficial to explore the potential sources of this heterogeneity further, such as differences in population characteristics, study designs, or diagnostic criteria for insomnia and dementia.

Subgroup Analyses: The authors have conducted subgroup analyses based on continent, follow-up years, gender, and insomnia subtypes. While these analyses provide valuable insights, they also reveal substantial differences in the risk estimates across different subgroups. For example, the risk of dementia associated with insomnia is markedly higher in Asian populations compared to European and American populations. These differences may be due to a variety of factors, including genetic predisposition, cultural differences, or variations in healthcare systems. The manuscript would benefit from a more in-depth discussion of these potential explanations and their implications for the generalizability of the findings.

3. Statistical Analysis: The statistical analysis in the manuscript appears to be performed appropriately and rigorously. The authors have used both fixed-effects and random-effects models to account for heterogeneity, and they have conducted sensitivity analyses to validate the robustness of their findings. The use of Egger’s test and Begg’s test to evaluate publication bias is appropriate, and the results suggest that publication bias is not a major concern in this meta-analysis.

However, the significant heterogeneity identified in some analyses remains a concern. The use of random-effects models is justified given the heterogeneity, but it also means that the pooled estimates should be interpreted with caution. The authors may want to consider conducting additional analyses to explore the sources of heterogeneity, such as meta-regression or stratified analyses based on study characteristics.

4. Interpretation of Findings: The interpretation of the findings is generally sound, and the authors have provided a thorough discussion of the potential mechanisms underlying the association between insomnia and dementia. The discussion of amyloid-beta (Aβ) deposition, Tau protein abnormalities, and the role of inflammation in the pathogenesis of dementia is well-supported by the existing literature. The manuscript also highlights the complex bidirectional relationship between sleep disorders and neurodegeneration, which is an important consideration.

However, the manuscript could be strengthened by a more detailed discussion of the limitations of the study and the potential impact of confounding factors. For example, many of the included studies relied on self-reported questionnaires to diagnose insomnia, which may introduce recall bias or misclassification. Additionally, while the authors have adjusted for several confounders, there may be residual confounding that could influence the results. A more explicit acknowledgment of these limitations would add transparency to the manuscript and provide a more balanced interpretation of the findings.

5. Ethical Considerations: The manuscript includes an ethics statement indicating that the data were analyzed anonymously and that the datasets used during the study are available from public databases. This is consistent with ethical guidelines for meta-analyses, and no major ethical concerns are noted.

6. Additional Comments:

The manuscript could benefit from a more detailed explanation of the search strategy, including the specific keywords and search terms used, as well as any limitations in the search process. This would allow readers to better assess the comprehensiveness of the literature search.

The authors have provided a clear and concise abstract, but it would be helpful to include a brief mention of the limitations of the study in the abstract as well, to provide a more balanced summary of the findings.

The manuscript is generally well-written, but there are a few minor grammatical errors and typos that should be corrected in the final version.

7. Conclusion: In conclusion, the manuscript presents a valuable contribution to the literature on the relationship between insomnia and dementia risk. The findings are supported by a rigorous and systematic approach to data synthesis, although the significant heterogeneity observed in some analyses warrants caution in the interpretation of the results. With some revisions, the manuscript has the potential to make a meaningful impact on the understanding of sleep disorders and neurodegenerative diseases.

I recommend that the authors address the concerns regarding heterogeneity and provide a more detailed discussion of the study's limitations. With these revisions, the manuscript would be suitable for publication.

6. PLOS authors have the option to publish the peer review history of their article (what does this mean? ). If published, this will include your full peer review and any attached files.

**Do you want your identity to be public for this peer review?** For information about this choice, including consent withdrawal, please see our Privacy Policy .

Reviewer #1: No

Reviewer #2: **Yes: ** Ankit Chalia

---

## [Author Response · Author response to Decision Letter 1]

19 Sep 2024

Dear reviewer, we have responded point by point to your questions in the file “response to reviewers”, and all the changes have been marked in red in the file “revised manuscript with track changes”.

---

## [Decision Letter · Decision Letter 1]

22 Jan 2025

Insomnia and Risk of All-Cause Dementia: A Systematic Review and Meta-Analysis

PONE-D-24-20907R1

Dear Dr.  Xiaoming

We’re pleased to inform you that your manuscript has been judged scientifically suitable for publication and will be formally accepted for publication once it meets all outstanding technical requirements.

Kind regards,

Frances Chung, M.B.B.S, F.R.C.P.C

Academic Editor

PLOS ONE

Additional Editor Comments (optional):

Reviewers' comments:

Reviewer's Responses to Questions

**Comments to the Author**

1. If the authors have adequately addressed your comments raised in a previous round of review and you feel that this manuscript is now acceptable for publication, you may indicate that here to bypass the “Comments to the Author” section, enter your conflict of interest statement in the “Confidential to Editor” section, and submit your "Accept" recommendation.

Reviewer #1: All comments have been addressed

Reviewer #2: All comments have been addressed

2. Is the manuscript technically sound, and do the data support the conclusions?

Reviewer #1: Yes

Reviewer #2: Yes

3. Has the statistical analysis been performed appropriately and rigorously? 

Reviewer #1: Yes

Reviewer #2: Yes

4. Have the authors made all data underlying the findings in their manuscript fully available?

Reviewer #1: Yes

Reviewer #2: Yes

5. Is the manuscript presented in an intelligible fashion and written in standard English?

Reviewer #1: Yes

Reviewer #2: Yes

6. Review Comments to the Author

Reviewer #1: (No Response)

Reviewer #2: I would like to commend the authors on their thorough and thoughtful revisions to the manuscript titled "Insomnia and Risk of All-Cause Dementia: A Systematic Review and Meta-Analysis." The improvements made have significantly enhanced the clarity, rigor, and overall quality of the manuscript. Below are some specific areas where the changes have been particularly effective:

Abstract: The abstract is now much more concise, and the emphasis on the practical implications of the findings related to early intervention for insomnia is a strong addition. This enhances the impact and relevance of the research, making it more accessible to a broad audience.

Introduction: The introduction has been refocused to highlight the importance of modifiable risk factors such as insomnia. The revised language is clearer and flows better, making the rationale for the study more compelling and easy to follow.

Methods: The detailed inclusion of the search strategy and explicit description of the keywords used significantly improve transparency. The additional clarification on how conflicts in study selection were resolved adds rigor to the methodological approach.

Results: The results section is now much clearer, particularly the subgroup analyses. The presentation of the data is more concise, and the additional context provided for the figures enhances the reader’s understanding of the key findings.

Discussion: The expanded discussion on heterogeneity provides important context for the study's findings and demonstrates a nuanced understanding of the variability in the included studies. The additional acknowledgment of study limitations, such as the use of self-reported data, further strengthens the discussion.

Conclusion: The revised conclusion does an excellent job of emphasizing the practical applications of the findings, particularly the potential for insomnia interventions to reduce dementia risk. This is a strong and clinically relevant takeaway.

Overall, the revisions have greatly improved the manuscript, and I appreciate the authors' dedication to addressing the feedback provided. The manuscript is now clearer, more concise, and better positioned to make a meaningful contribution to the field. Excellent work!

7. PLOS authors have the option to publish the peer review history of their article (what does this mean? ). If published, this will include your full peer review and any attached files.

**Do you want your identity to be public for this peer review?** For information about this choice, including consent withdrawal, please see our Privacy Policy .

Reviewer #1: No

Reviewer #2: No

---

## [Editor Report · Acceptance letter]

PONE-D-24-20907R1

PLOS ONE

Dear Dr. shen,

I'm pleased to inform you that your manuscript has been deemed suitable for publication in PLOS ONE. Congratulations! Your manuscript is now being handed over to our production team.

Kind regards,

on behalf of

Dr. Frances Chung

Academic Editor

PLOS ONE